# FEW-SHOT BAYESIAN OPTIMIZATION WITH DEEP KERNEL SURROGATES

**Martin Wistuba**
IBM Research
Dublin, Ireland
`martin.wistuba@ibm.com`

**Josif Grabocka**
University of Freiburg
Freiburg, Germany
`grabocka@cs.uni-freiburg.de`

## ABSTRACT

Hyperparameter optimization (HPO) is a central pillar in the automation of machine learning solutions and is mainly performed via Bayesian optimization, where a parametric surrogate is learned to approximate the black box response function (e.g. validation error). Unfortunately, evaluating the response function is computationally intensive. As a remedy, earlier work emphasizes the need for *transfer learning* surrogates which learn to optimize hyperparameters for an algorithm from other tasks. In contrast to previous work, we propose to rethink HPO as a *few-shot learning* problem in which we train a shared deep surrogate model to quickly adapt (with few response evaluations) to the response function of a new task. We propose the use of a deep kernel network for a Gaussian process surrogate that is meta-learned in an end-to-end fashion in order to jointly approximate the response functions of a collection of training data sets. As a result, the novel few-shot optimization of our deep kernel surrogate leads to new state-of-the-art results at HPO compared to several recent methods on diverse metadata sets.

## 1 INTRODUCTION

Many machine learning models have very sensitive hyperparameters that must be carefully tuned for efficient use. Unfortunately, finding the right setting is a tedious trial and error process that requires expert knowledge. AutoML methods address this problem by providing tools to automate hyperparameter optimization, where Bayesian optimization has become the standard for this task (Snoek et al., 2012). It treats the problem of hyperparameter optimization as a black box optimization problem. Here the black box function is the hyperparameter response function, which maps a hyperparameter setting to the validation loss. Bayesian optimization consists of two parts. First, a surrogate model, often a Gaussian process, is used to approximate the response function. Second, an acquisition function is used that balances the trade-off between exploration and exploitation. In a sequential process, hyperparameter settings are selected and evaluated, followed by an update of the surrogate model.

Recently, several attempts have been made to extend Bayesian optimization to account for a transfer learning setup. It is assumed here that historical information on machine learning algorithms is available with different hyperparameters. This can either be because this information is publicly available (e.g. OpenML) or because the algorithm is repeatedly optimized for different data sets. To this end, several transfer learning surrogates have been proposed that use this additional information to reduce the convergence time of Bayesian optimization.

We propose a new paradigm for accomplishing the knowledge transfer by reconceptualizing the process as a few-shot learning task. Inspiration is drawn from the fact that there are a limited number of black box function evaluations for a new hyperparameter optimization task (i.e. few shots) but there are ample evaluations of related black box objectives (i.e. evaluated hyperparameters on other data sets). This approach has several advantages. First, a single model is learned that is trained to quickly adapt to a new task when few examples are available. This is exactly the challenge we face when optimizing hyperparameters. Second, this method can be scaled very well for any number of considered tasks. This not only enables the learning from large metadata sets but also enables the problem of label normalization to be dealt with in a new way. Finally, we present an

evolutionary algorithm that can use surrogate models to get a warm start initialization for Bayesian optimization. All of our contributions are empirically compared with several competitive methods in three different problems. Two ablation studies provide information about the influence of the individual components. Concluding, our contributions in this work are:

- This is the first work that, in the context of hyperparameter optimization (HPO), trains the initialization of the parameters of a probabilistic surrogate model from a collection of meta-tasks by few-shot learning and then transfers it by fine-tuning the initialized kernel parameters to a target task.

- We are the first to consider transfer learning in HPO as a few-shot learning task.

- We set the new state of the art in transfer learning for the HPO and provide ample evidence that we outperform strong baselines published at ICLR and NeurIPS with a statistically significant margin.

- We present an evolutionary algorithm that can use surrogate models to get a warm start initialization for Bayesian optimization.

## 2 RELATED WORK

The idea of using transfer learning to improve Bayesian optimization is being investigated in several papers. The early work suggests learning a single Gaussian process for the entire data (Bardenet et al., 2013; Swersky et al., 2013; Yogatama & Mann, 2014). Since the training of a Gaussian process is cubic in the number of training points, this idea does not scale well. This problem has recently been addressed by several proposals to use ensembles of Gaussian processes where a Gaussian process is learned for each task (Wistuba et al., 2018; Feurer et al., 2018). This idea scales linearly in the number of tasks but still cubically in the number of training points per task. Thus, the problem persists in scenarios where there is a lot of data available for each task.

Bayesian neural networks are a possible more scalable way of learning with large amounts of data. For example, Schilling et al. (2015) propose to use a neural network with task embedding and variable interactions. To obtain mean and variance predictions, the authors propose using an ensemble of models. In contrast, Springenberg et al. (2016) use a Bayesian multi-task neural network. However, since training Bayesian neural networks is computationally intensive, Perrone et al. (2018) propose a more scalable approach. They suggest using a neural network that is shared by all tasks and using Bayesian linear regression for each task. The parameters are trained jointly on the entire data. While our work shares some similarities with the previous work, our algorithm has unique properties. First of all, a meta-learning algorithm is used, which is motivated by recent work on model-agnostic meta-learning for few-shot learning (Finn et al., 2017). This will allow us to integrate all task-specific parameters out such that the model does not grow with the number of tasks. As a result, our algorithm scales very well with the number of tasks. Second, while we are also using a neural network, we combine it with a Gaussian process with a nonlinear kernel in order to obtain uncertainty predictions.

A simpler solution for using the transfer learning idea in Bayesian optimization is initialization (Feurer et al., 2015; Wistuba et al., 2015a). The standard Bayesian optimization routine with a simple Gaussian process is used in this case but it is warm-started by a number of hyperparameter settings that work well for related tasks. We also explore this idea in the context of this paper by proposing a simple evolutionary algorithm that can use a surrogate model to estimate a data-driven warm start initialization sequence. The use of an evolutionary algorithm is motivated by its ease of implementation and natural capability to deal with continuous and discrete hyperparameters.

## 3 PRELIMINARIES

### 3.1 BAYESIAN OPTIMIZATION

Bayesian optimization is an optimization method for computationally intensive black box functions that consists of two main components, the surrogate model and the acquisition function. The surrogate model is a probabilistic model with mean $\mu$ and variance $\sigma^2$, which tries to approximate the unknown black box function $f$, in the following also *response function*. The acquisition function

can provide a score for each feasible solution based on the prediction of the surrogate model. This score balances between exploration and exploitation. The following steps are carried out sequentially. First, the feasible solution that maximizes the acquisition function is evaluated. In this way a new observation $(\mathbf{x}, f(\mathbf{x}))$ is obtained. Then, the surrogate model is updated based on the entirety of all observations $\mathcal{D}$. This sequence of steps is carried out until a previously defined convergence criterion occurs (e.g. exhaustion of the time budget). In Figure 1 we provide an example how Bayesian optimization is used to maximize a sine wave.

Expected Improvement (Jones et al., 1998) is one of the most commonly used acquisition functions and will also be used in all our experiments. It is defined as

$$a(\mathbf{x}|\mathcal{D}) = \mathbb{E}\left[\max\left\{f(\mathbf{x}) - y_{\max}, 0\right\}\right] , \tag{1}$$

where $y_{\max}$ is the largest observed value of $f$.

## 3.2 Gaussian Processes

We have a training set $\mathcal{D}$ of $n$ observations, $\mathcal{D} = \{(\mathbf{x}_i, y_i)|i = 1, \ldots, n\}$, where $y_i$ are noisy observations of the function values $y_i = f(\mathbf{x}_i) + \varepsilon$. We assume that the noise $\varepsilon$ is additive independent identically distributed Gaussian with variance $\sigma_n^2$. For Gaussian processes (Rasmussen & Williams, 2006) we consider $y_i$ to be a random variable and the joint distribution of all $y_i$ is assumed to be multivariate Gaussian distributed:

$$\mathbf{y} \sim \mathcal{N}\left(m\left(\mathbf{X}\right), k\left(\mathbf{X}, \mathbf{X}\right)\right) . \tag{2}$$

A Gaussian process is completely specified by its mean function $m$, its covariance function $k$, and may depend on parameters $\boldsymbol{\theta}$. A common choice is to set $m(\mathbf{x}_i) = 0$. At inference time for instances $\mathbf{x}_*$, the assumption is that their ground truth $\mathbf{f}_*$ is jointly Gaussian with $\mathbf{y}$

$$\left[\begin{array}{c} \mathbf{y} \\ \mathbf{f}_* \end{array}\right] \sim \mathcal{N}\left(\mathbf{0}, \left(\begin{array}{cc} \mathbf{K}_n & \mathbf{K}_* \\ \mathbf{K}_*^T & \mathbf{K}_{**} \end{array}\right)\right) , \tag{3}$$

where

$$\mathbf{K}_n = k\left(\mathbf{X}, \mathbf{X}|\boldsymbol{\theta}\right) + \sigma_n^2\mathbf{I}, \ \mathbf{K}_* = k\left(\mathbf{X}, \mathbf{X}_*|\boldsymbol{\theta}\right), \ \mathbf{K}_{**} = k\left(\mathbf{X}_*, \mathbf{X}_*|\boldsymbol{\theta}\right) \tag{4}$$

for brevity. Then, the posterior predictive distribution has mean and covariance

$$\mathbb{E}\left[\mathbf{f}_*|\mathbf{X}, \mathbf{y}, \mathbf{X}_*\right] = \mathbf{K}_*^T\mathbf{K}_n^{-1}\mathbf{y}, \ \text{cov}\left[\mathbf{f}_*|\mathbf{X}, \mathbf{X}_*\right] = \mathbf{K}_{**} - \mathbf{K}_*^T\mathbf{K}_n^{-1}\mathbf{K}_* \tag{5}$$

Examples for covariance functions are the linear or squared exponential kernel. However, these kernel functions are designed by hand. The idea of *deep kernel learning* (Wilson et al., 2016) is to learn the kernel function. They propose to use a neural network $\varphi$ to transform the input $\mathbf{x}$ to a latent representation which serves as input for the kernel function.

$$k_{\text{deep}}(\mathbf{x}, \mathbf{x}'|\boldsymbol{\theta}, \mathbf{w}) = k(\varphi(\mathbf{x}, \mathbf{w}), \varphi(\mathbf{x}', \mathbf{w})|\boldsymbol{\theta}) \tag{6}$$

## 4 Few-Shot Bayesian Optimization

Hyperparameter optimization is traditionally either tackled as an optimization problem without prior knowledge about the response function or alternatively as a multi-task or transfer learning problem. In the former, every search basically starts from scratch. In the latter, one or multiple models are trained that attempt to reuse knowledge of the source tasks for the target task.

In this work we will address the problem as a few-shot problem. Given $T$ related source tasks and very few examples of the target task, we want to make reliable predictions. For each of the source tasks we have observations $\mathcal{D}^{(t)} = \{(\mathbf{x}_i^{(t)}, y_i^{(t)})\}_{i=1\ldots n^{(t)}}$, where $\mathbf{x}_i^{(t)}$ is a hyperparameter setting and $y_i^{(t)}$ is the noisy observations of $f^{(t)}(\mathbf{x}_i^{(t)})$, i.e. a validation score of a machine learning model on data set $t$. In the following we will denote the set of all data points by $\mathcal{D} := \bigcup_{t=1}^T \mathcal{D}^{(t)}$. Model-agnostic meta-learning (Finn et al., 2017) has become a popular choice for few-shot learning and we propose to use an adaptation for Gaussian processes (Patacchiola et al., 2020) as a surrogate model within the Bayesian optimization framework. The idea is simple. A deep kernel $\varphi$ is used to learn parameters across tasks such that all its parameters $\boldsymbol{\theta}$ and $\mathbf{w}$ are task-independent. All

task-dependent parameters are kept separate which allows to marginalize its corresponding variable out when solely optimizing for the task-independent parameters. If we assume that the posteriors over $\boldsymbol{\theta}$ and $\mathbf{w}$ are dominated by their respective maximum likelihood estimates $\hat{\boldsymbol{\theta}}$ and $\hat{\mathbf{w}}$, we can approximate the posterior predictive distribution by

$$p\left(f_*|\mathbf{x}_*,\mathcal{D}\right) = \int p\left(f_*|\mathbf{x}_*,\boldsymbol{\theta},\mathbf{w}\right) p\left(\boldsymbol{\theta},\mathbf{w}|\mathcal{D}\right) \mathrm{d}\boldsymbol{\theta},\mathbf{w} \approx p\left(f_*|\mathbf{x}_*,\mathcal{D},\hat{\boldsymbol{\theta}},\hat{\mathbf{w}}\right) \quad (7)$$

We estimate $\hat{\boldsymbol{\theta}}$ and $\hat{\mathbf{w}}$ by maximizing the log marginal likelihood for all tasks.

$$\log p\left(\mathbf{y}^{(1)},\ldots,\mathbf{y}^{(T)}|\mathbf{X}^{(1)},\ldots,\mathbf{X}^{(T)},\boldsymbol{\theta},\mathbf{w}\right) = \sum_{t=1}^{T} \log p\left(\mathbf{y}^{(t)}|\mathbf{X}^{(t)},\boldsymbol{\theta},\mathbf{w}\right) \quad (8)$$

$$\propto -\sum_{t=1}^{T} \left(\mathbf{y}^{(t)\mathrm{T}}\mathbf{K}_n^{(t)^{-1}}\mathbf{y}^{(t)} + \log\left|\mathbf{K}_n^{(t)}\right|\right) \quad (9)$$

We maximize the marginal likelihood with stochastic gradient ascent (SGA). Each batch contains data for one task. It is important to note that this batch data does not correspond to the data in $\mathcal{D}^{(t)}$, but rather a subset of it. With small batch sizes, this is an efficient way to train the Gaussian process, regardless of how many tasks knowledge is transferred from or how large $\mathcal{D}^{(t)}$ is. It is important to note that there is a correlation between all data per task and not just one batch. Hence the stochastic gradient is a biased estimator of the full gradient. For a long time we lacked the theoretical understanding of how SGA would behave in this situation. Fortunately, recent work (Chen et al., 2020) proved that SGA still successfully converges and restores model parameters in these cases. The results by Chen et al. (2020) guarantee a $\mathcal{O}(1/K)$ optimization margin of error, where $K$ is the number of iterations.

The training works as follows. In each iteration, a task $t$ is sampled uniformly at random from all $T$ tasks. Then we sample a batch of training instances uniformly at random from $\mathcal{D}^{(t)}$. Finally, we calculate the log marginal likelihood for this batch (Equation 8) and update the kernel parameters with one step in the direction of the gradient.

In hyperparameter optimization, we are only interested in the hyperparameter setting that works best according to a predefined metric of interest. Therefore, only the ranking of the hyperparameter settings is important, not the actual value of the metric. Therefore, we are interested in a surrogate model whose prediction strongly correlate with the response function and the squared error is of little to no interest for us. In practice, however, the minimum / maximum score and the range of values differ significantly between the tasks which makes it challenging to obtain a strongly correlating surrogate model. The most common way to address this problem is to normalize the labels, e.g. by z-normalizing or scaling the labels to $[0,1]$ per data set. However, this does not fully solve the problem. The main problem is that this label normalization must also be applied to the target task. This is not possible with a satisfactory degree of accuracy, especially when only a few examples are available, since the approximate values for minimum / maximum or mean / standard deviation are insufficiently accurate.

Since our proposed method scales easily to any number of tasks, we can afford to consider another option. We propose a task augmentation strategy that addresses the label normalization issue by randomly scaling the labels for each batch. Since $y_{\min}$ and $y_{\max}$ are the minimum and maximum values for all $T$ tasks, we can generate augmented tasks for each batch $\mathcal{B} = \{\mathbf{x}_i, y_i\}_{i=1,\ldots,b} \sim \mathcal{D}^{(t)}$, where $b$ is the batch size, as follows. A lower and upper limit is sampled for each sample batch,

$$l \sim \mathcal{U}(y_{\min}, y_{\max}), u \sim \mathcal{U}(y_{\min}, y_{\max}) , \quad (10)$$

such that $l < u$ holds. Then the labels for that batch are scaled to this range,

$$\mathbf{y} \leftarrow \frac{\mathbf{y} - l}{u - l} . \quad (11)$$

No further changes are required. We summarize this procedure in Algorithm 1. The idea here is to learn a representation that is invariant with respect to various offsets and ranges so that the target task does not require normalization. This strategy is not possible for other hyperparameter optimization methods, since this either increases the training data set size even further and thus becomes

---

**Algorithm 1:** Few-Shot GP Surrogate

---

**Input:** Learning rates $\alpha$ and $\beta$, training data $\mathcal{D}$, kernel $k$, and neural network $\varphi$.

**while** *not done* **do**

    Sample task $t \sim \mathcal{U}(\{1, \ldots, T\})$;

    Estimate $l$ and $u$ (Equation 10);

    **for** $b_n$ *times* **do**

        Sample batch $\mathcal{B} = \{(\mathbf{x}_i, y_i)\}_{i=1,\ldots,b} \sim \mathcal{D}^{(t)}$ and scale labels $\mathbf{y}$ using $l$ and $u$;

        Compute marginal likelihood $\mathcal{L}$ on $\mathcal{B}$. (Equation 8);

        $\boldsymbol{\theta} \leftarrow \boldsymbol{\theta} + \alpha \nabla_{\boldsymbol{\theta}} \mathcal{L}$;

        $\mathbf{w} \leftarrow \mathbf{w} + \beta \nabla_{\mathbf{w}} \mathcal{L}$;

    **end**

**end**

---

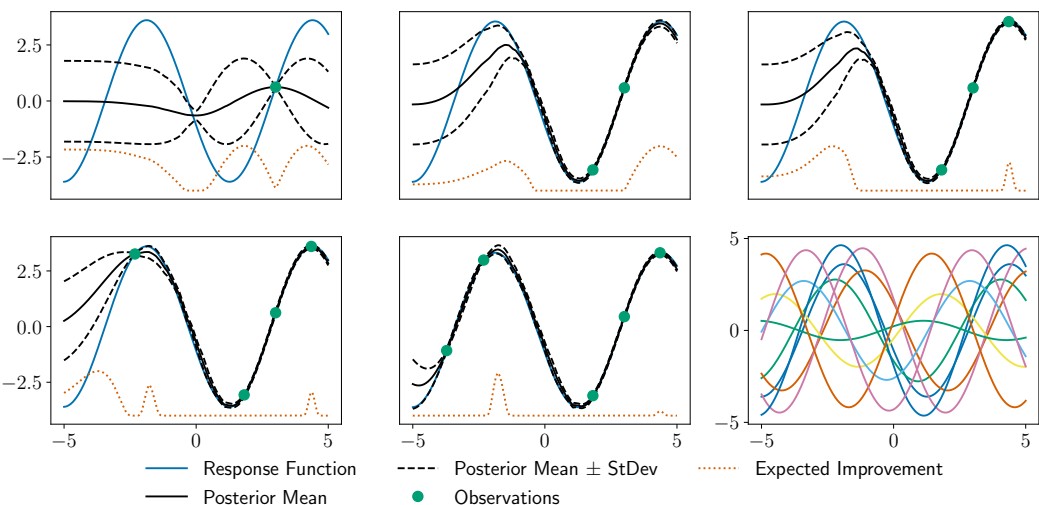

Figure 1: Demonstration of five steps of Bayesian Optimization with FSBO for maximizing a sine wave (blue). One maximum is discovered within only three steps. Expected Improvement has been scaled to improve readability. In black the predictions of the surrogate model. Bottom right are examples of the source tasks. The deep kernel consists of a spectral kernel (Wilson & Adams, 2013) combined with a two-layer neural network ($1 \rightarrow 64 \rightarrow 64$).

computationally impossible (Bardenet et al., 2013; Swersky et al., 2013; Yogatama & Mann, 2014) or thousands of new tasks would have to be generated, which is also is not practical (Springenberg et al., 2016; Perrone et al., 2018; Wistuba et al., 2018; Feurer et al., 2018).

At test time, the posterior predictive distribution or the target task $T + 1$ is computed according to Equation 5. In practice, the target data set can be very different to the learned prior. For this reason, the deep kernel parameters are fine-tuned on the target task's data for few epochs. Our task augmentation strategy described earlier has resulted in a model that has become invariant for different scales of $\mathbf{y}$. For this reason we do not apply this strategy to the target task $T + 1$ and only use it for all source tasks 1 to $T$. We discuss the usefulness of this step in more detail when we discuss the empirical results.

## 5   A MOTIVATING EXAMPLE

We would like to first motivate our method, FSBO, with an example, similar to the one described in Nichol et al. (2018). The aim is to find the argument $x$ that maximizes a one-dimensional sine wave of the form $f^{(t)}(x) = a^{(t)} \sin(x + b^{(t)})$. It is not known that the function is a sine wave and that one only has to determine the amplitude $a$ and phase $b$. However, information about other sine waves

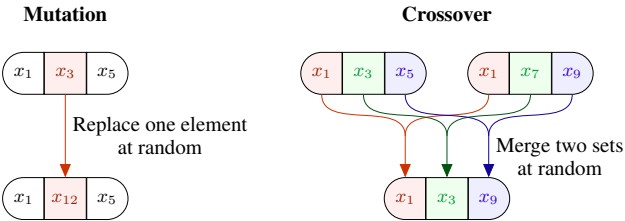

Figure 2: Examples for the mutation and crossover operation with $I = 3$.

is available to the optimizer which are considered to be related. These source tasks $t = 1, \ldots, T$ are generated randomly with $a \sim \mathcal{U}(0.1, 5)$ and $b \sim \mathcal{U}(0, 2\pi)$. 50 examples $(x_i^{(t)}, f^{(t)}(x_i^{(t)}))$ are available for each task $t$, whereby the points $x_i^{(t)} \in [-5, 5]$ are evenly spaced.

It should be noted at this point that the expected value for each $x_i$ is 0. Thus, training on the data described above leads to a model predicting 0 for every $x_i$. However, the used meta-learning procedure allows for accurately reconstructing the underlying sine wave after only a few examples (see Figure 1). Provided that the response function of the original task is similar to the target task, this is a very promising technique for accelerating the search for good hyperparameter settings.

## 6 A DATA-DRIVEN INITIALIZATION

The proposed few-shot surrogate model requires only a few examples before reliable predictions can be made for the target data set. How do you choose the initial hyper parameter settings? The simplest idea would be to pick them at random or use Latin Hypercube Sampling (LHS) (McKay et al., 1979). Since we already have data from various tasks and a surrogate model that can impute missing values, we propose a data-driven warm start approach. If we evaluate the performance of a hyperparameter optimization algorithm with a loss function $\mathcal{L}$, we use an evolutionary algorithm to find a set with $I$ hyperparameter settings which minimizes this loss on the source tasks.

$$\mathbf{X}^{(\text{init})} = \underset{\mathbf{X} \in \mathcal{X}^I}{\arg\min} \sum_{t=1}^{T} \mathcal{L}\left(f^{(t)}, \mathbf{X}\right) = \underset{\mathbf{X} \in \mathcal{X}^I}{\arg\min} \sum_{t=1}^{T} \min_{\mathbf{x} \in \mathbf{X}} \tilde{f}^{(t)}(\mathbf{x}) \tag{12}$$

The loss of a set of hyperparameters depends on the response function values for each of the elements. Since these are not available for every $\mathbf{x}$, we approximate $f^{(t)}(\mathbf{x})$ with the prediction of the surrogate model described in the last section whenever this is necessary.

Arbitrary loss function can be considered. The specific loss function used in our experiments is the normalized regret and is defined on the right side of Equation 12, where $\tilde{f}^{(t)}$ is derived from $f^{(t)}$ by scaling it to $[0, 1]$ range:

$$\tilde{f}^{(t)}(\mathbf{x}) = \frac{f^{(t)}(\mathbf{x}) - f_{\min}^{(t)}}{f_{\max}^{(t)} - f_{\min}^{(t)}}, \tag{13}$$

where $f_{\max}^{(t)}$ and $f_{\min}^{(t)}$ are the maximum and minimum of $f^{(t)}$, respectively.

The evolutionary algorithm works as follows. We initialize the population with sets containing $I$ random settings, the settings being sampled in proportion to their performance according to the predictions. Precisely, a setting $\mathbf{x}$ is sampled proportional to

$$\exp\left(-\min_{t \in \{1, \ldots, T\}} \tilde{f}^{(t)}(\mathbf{x})\right). \tag{14}$$

The best sets are then selected to be refined using an evolutionary algorithm. The algorithm randomly chooses either to mutate a set or to perform a crossover operation between two sets. When mutating a set, a setting is removed uniformly at random and a new setting is added proportional to its predicted performance (Figure 2, left). The crossover operation creates a new set and adds elements from the union of the parent sets until the new set has $I$ settings (Figure 2, right). The new

set is added to the population. After 100,000 steps, the algorithm is stopped and the best set is used as the set of initial hyperparameter settings. In Figure 4 we compare this warm start initialization with the simple random or LHS initialization.

# 7    EXPERIMENTS

We used three different optimization problems to compare the different hyperparameter optimization methods: AdaBoost, GLMNet, and SVM. We created the GLMNet and SVM metadata set by downloading the 30 data sets with the most reported hyperparameter settings from OpenML for each problem. The AdaBoost data set is publicly available (Wistuba et al., 2018). The number of settings per data set varies, the number of settings across all tasks for the GLMNet problem is approximately 800,000. See the appendix for more details about the metadata sets. We compare the following list of hyperparameter optimization methods.

**Random Search** This is a simple but strong baseline for hyperparameter optimization (Bergstra & Bengio, 2012). Hyperparameters settings are selected uniformly at random from the search space.

**Gaussian Process (GP)** Bayesian optimization with a Gaussian process as a surrogate model is a standard and strong baseline (Snoek et al., 2012). We use a Matérn 5/2 kernel with ARD and rely on the scikit-optimize implementation. We compare with two variations. The first is the vanilla method, which uses Latin Hypercube Sampling (LHS) with design size of 10 to initialize. The second method uses the warm start (WS) method described in Section 6 to find an initial set of 5 settings to use the knowledge from other data sets.

**RGPE** RGPE (Feurer et al., 2018) is one of the latest examples of methods that use GP ensembles to transfer knowledge across task (Wistuba et al., 2016; Schilling et al., 2016; Wistuba et al., 2018). In this case a GP is trained for each individual task and in a final step the predictions of the GPs are aggregated. These ensembles scale linearly in the number of tasks, but the computation time is still cubic and the space requirement is still quadratic in the number of data points per task. For this reason we can only present results for AdaBoost and not for the other optimization problems, which have significantly more data points.

**MetaBO** Using the acquisition function instead of the surrogate model for transfer learning is another option. MetaBO is the state-of-the-art transfer acquisition function and was presented last year at ICLR (Volpp et al., 2020). We use the implementation provided by the authors.

**ABLR** The state-of-the-art surrogate model for scalable hyperparameter transfer learning (Perrone et al., 2018). This method uses a multi-task GP which combines a linear kernel with a neural network and scales to very large data sets. Transfer learning is achieved by sharing the neural network parameters across different tasks. By ABLR (WS) we are referring to a version of ABLR that uses the same warm start as FSBO.

**Multi-Head GPs** This closely resembles ABLR but uses the same deep kernel as our proposed method. The main difference from our proposed method is that it is using a GP for every task, only shares the neural network across tasks, and uses standard stochastic gradient ascent.

**Few-Shot Bayesian Optimization (FSBO)** Our proposed method described in Section 4. The deep kernel is composed of a two-layer neural network ($128 \rightarrow 128$) with ReLU activations and a squared-exponential kernel. We use the Adam optimizer with learning rate $10^{-3}$ and a batch size of fifty. The warm start length is five.

The experiments are repeated ten times and evaluated in a leave-one-task-out cross-validation. This means that all transfer learning methods use one task as the target task and all other tasks as source tasks. For AdaBoost we use the same train/test split as used by Volpp et al. (2020) instead. We report the aggregated results for all tasks within one problem class with respect to the mean of normalized regrets. The normalized regret for a task is obtained by first scaling the response function values between 0 and 1 before calculating the regret (Wistuba et al., 2018), i.e.

$$\tilde{r}^{(t)} = \tilde{f}^{(t)}(\mathbf{x}_{\min}) - \tilde{f}^{(t)}_{\min}, \tag{15}$$

| Method | AdaBoost | | | GLMNet | | | SVM | | |
|---|---|---|---|---|---|---|---|---|---|
| | 15 | 33 | 50 | 33 | 67 | 100 | 33 | 67 | 100 |
| RANDOM | *4.87* | 3.02 | 2.16 | 0.85 | 0.40 | 0.29 | 2.01 | 1.12 | 0.83 |
| GP (LHS) | 3.87 | 2.49 | 2.23 | 1.32 | 1.01 | 0.73 | 1.94 | 1.40 | 1.14 |
| GP (WS) | *3.25* | *1.66* | *1.02* | *0.86* | 0.36 | 0.24 | *1.10* | 0.81 | 0.65 |
| RGPE | 5.29 | 3.26 | 2.83 | _[1] | _[1] | _[1] | _[1] | _[1] | _[1] |
| METABO | 5.27 | 3.52 | 1.96 | 11.02 | 10.97 | 10.96 | 12.39 | 12.39 | 11.93 |
| ABLR | *4.56* | 1.44 | 1.24 | 1.77 | 0.50 | 0.40 | 1.81 | 1.23 | 0.84 |
| ABLR (WS) | *3.17* | *1.72* | *1.17* | 0.54 | 0.36 | 0.29 | 1.47 | 1.08 | 0.87 |
| MULTI-HEAD GPs (WS) | 6.78 | 3.45 | 1.80 | 2.83 | 2.80 | 2.68 | 11.20 | 9.82 | 8.72 |
| FSBO | **3.10** | **1.13** | **0.80** | **0.42** | **0.22** | **0.16** | **0.79** | **0.51** | **0.36** |

Table 1: FSBO obtains better results for all hyperparameter optimization problems. The best results are in **bold**. Results that are not significantly worse than the best are in *italics*. Used initialization in parentheses, (LHS) - Latin Hypercube Sampling, (WS) - Warm Start.

where $\tilde{f}^{(t)}$ is the normalized response function (Equation 13), $\tilde{f}_{\min}^{(t)}$ is the global minimum for task $t$, and $\mathbf{x}_{\min}$ is the best discovered hyperparameter setting by the optimizer.

## 7.1 HYPERPARAMETER OPTIMIZATION

We conduct experiments on three different metadata sets and report the aggregated mean of normalized regrets in Table 1. The best results are in bold. Results that are not significantly worse than the best are in italics. We determined the significance using the Wilcoxon signed rank test with a confidence level of 95%. Results are reported after every 33 trials of Bayesian optimization for GLMNet and SVM. Since AdaBoost has fewer test examples, we report its results in shorter intervals.

Our proposed FSBO method outperforms all other transfer methods in all three tasks. Results are significantly better but in the case of AdaBoost where GP (WS) achieves similar but on average worse results. The results obtained for MetaBO are the worst. We contacted Volpp et al. (2020) and they confirmed that our results are obtained correctly. The problem is apparently that the Reinforcement Learning method does not work well for larger number of trials. We provide a longer discussion in the appendix. Also ABLR and the very related Multi-Head GP are not performing very well. As we show in the next section, one possible reason for this might be because the neural network parameters are fixed which will prohibit a possibly required adaptation to the target task. The vanilla GP and its transfer variant that uses a warm start turn out to be among the strongest baselines. This is in particular true for the warm start version which is the second best method. This simple baseline of knowledge transfer is often not taken into account when comparing transfer surrogates, although it is easy to implement. Due to the very competitive results, we recommend using it as a standard baseline to assess the usefulness of new transfer surrogates.

## 7.2 COMPONENT CONTRIBUTIONS

In this section we highlight the various components that make a significant difference to ABLR. The results for each setup are reported in Figure 3. The Multi-Head GP is a surrogate model that shares the neural network between tasks but uses a separate Gaussian process for each task. It closely resembles the idea of ABLR but is closer to our proposed implementation of FSBO. Starting from this configuration, we add various components that will eventually lead to our proposed model FSBO. We consider Multi-Head GP (WS), a version that additionally uses the warm start method described in Section 6 instead of a random initialization. Multi-Head GP (fine-tune) not only updates the kernel parameters when a new observation is received but also fine-tunes the parameters of the neural network. Finally, FSBO is our proposed method, which uses only one Gaussian process for all tasks. We see that all Multi-Head GP versions have a problem adapting to the target tasks efficiently. Fine-tuning the deep kernel is an important part of learning FSBO. Although FSBO outperforms all Multi-Head GP versions without fine-tuning, the additional use of it makes for a significant improvement. We analyzed the reason for this in more detail. We observed that the learned neural

---

[1]Out-of-memory exception due to too large data.

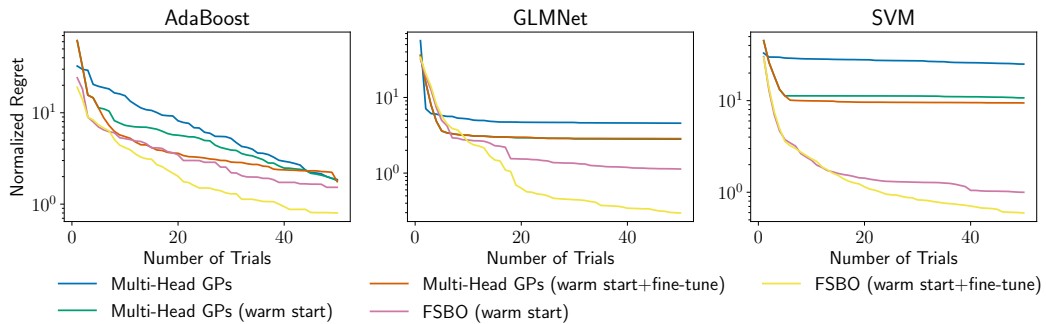

Figure 3: Comparison of the contribution of the various FSBO components to the final solution. Each component makes its own orthogonal contribution.

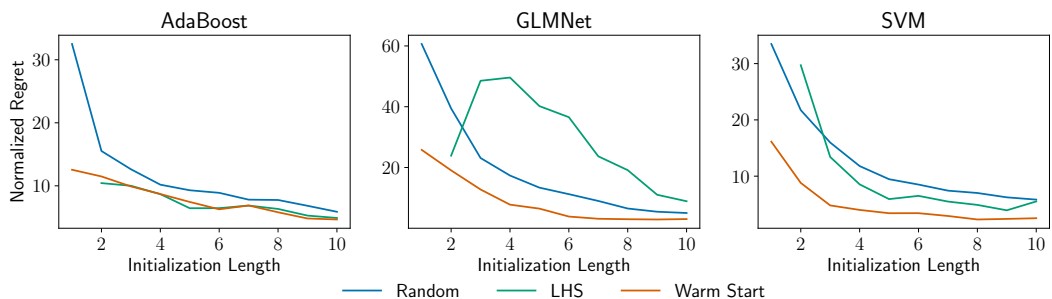

Figure 4: The warm start initialization yields the best results on GLMNet and SVM for all initialization lengths. For AdaBoost it is comparable to LHS.

network provides a strong prior that leads to strong exploitation behavior. Fine-tuning prevents this and thus ensures that the method does not get stuck in local optima.

### 7.3 Data-Driven Warm Start

The use of a warm start is not the main contribution of this paper but it is interesting to explore further nonetheless. First of all, for some methods this is the only differentiating factor. We compare our suggested warm start initialization with a random and a Latin hypercube sampling (LHS) initialization in Figure 4. Considering that GP (WS) always performed better than GP (LHS) in Table 1, one would expect that the warm start itself also performs better than LHS. While this is the case for GLMNet and SVM, there are no significant differences for the AdaBoost problem. Our assumption is that in this case the warm start might not have always found good settings as part of the initialization, it still explored areas in the search space close to the optimum. This would facilitate finding a better solution in one of the subsequent steps of Bayesian optimization.

## 8 Conclusions

In this work, we propose to rethink hyperparameter optimization as a few-shot learning problem in which we train a shared deep surrogate model to quickly adapt (with few response evaluations) to the response function of a new task. We propose the use of a deep kernel network for a Gaussian process surrogate that is meta-learned in an end-to-end fashion in order to jointly approximate the response functions of a collection of training data sets. This few-shot surrogate model is used for two different purposes. First, we use it in combination with an evolutionary algorithm in order to estimate a data-driven warm start initialization for Bayesian optimization. Second, we use it directly for Bayesian optimization. In our empirical evaluation on three hyperparameter optimization problems, we observe significantly better results than with state-of-the-art methods that use transfer learning.

ACKNOWLEDGMENTS

This work has been supported by European Union's Horizon 2020 research and innovation programme under grant number 951911 - AI4Media. Prof. Grabocka is thankful to the Eva Mayr-Stihl Foundation for their generous research grant.

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

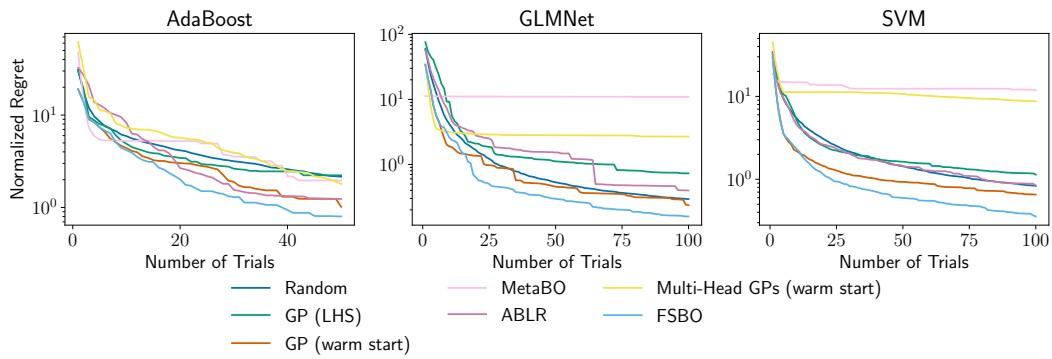

Figure 5: FSBO is the best of all considered methods.

## A   HYPERPARAMETER OPTIMIZATION

In the main paper we report the normalized regret after every 33 trials of Bayesian optimization in a table. This allows us to also report statistical significance. In Figure 5 we show results for all 100 trials. The conclusions remain unchanged. FSBO provides consistently the best results.

## B   CHALLENGES WITH METABO

The results reported in the main paper for MetaBO seem not to align with the numbers reported by Volpp et al. (2020), in particular for AdaBoost. In order to understand this behavior, we conducted a deeper analysis for the AdaBoost metadata set which was used in the evaluation of Volpp et al. (2020) as well. Using the authors' code[2], we executed the same scripts used by the authors to report their results for $T = 15$ trials. Furthermore, we changed two lines in the scripts to train MetaBO for $T = 50$, the number of trials considered for AdaBoost in our experiment. We report the outcome of these two experiments in Figure 6. In the two left plots we show how the learned policies for $T = 15$ and $T = 50$ perform on validation and test during training. The authors' default setting for the total number of iterations is 2000. However, we noticed a sudden drop in regret for the setting $T = 50$ around 2000 iterations. For that reason we assumed that further training might help to further improve MetaBO and increased it to 5000 iterations. As can be seen in the middle plot, that was not the case. We observe that the policy learned for $T = 15$ has a lower regret than the one for $T = 50$ even though it is only trying 15 compared to 50 trials. In the right plot we show the results compared to a simple random search baseline. The results for $T = 15$ are in line with those reported by Volpp et al. (2020). We contacted the authors and they confirmed that we use their code correctly. Their explanation is that the increase of episode length makes this a harder problem for Reinforcement Learning. This results in worse results for $T = 50$ and even worse for $T = 100$ in case of the GLMNet and SVM problem. They proposed to randomly vary $T$ between 5 and 50 during the training phase. We also considered this training protocol and report the results in Figure 6. This improves over training only with $T = 50$ but does not improve over a simple random search.

## C   METADATA

We created the GLMNet and SVM metadata using OpenML. We chose the 30 OpenML flows with the most observations. We also limited ourselves to the uploaded results from user with ID 2702 (OpenML_Bot R), who uploaded the majority of all runs, to ensure that the metadata was generated under similar circumstances. In some cases the choice of a single hyperparameter is not indicated. In such a case, we assume that the default value was used. The GLMNet metadata has two continous hyperparameters: the elastic-net mixing parameter $\alpha \in [0, 1]$ and the regularization parameter $\lambda \in [2^{-10}, 2^{10}]$. The SVM metadata has two mandatory hyperparameters and two conditional

---

[2]https://github.com/boschresearch/MetaBO

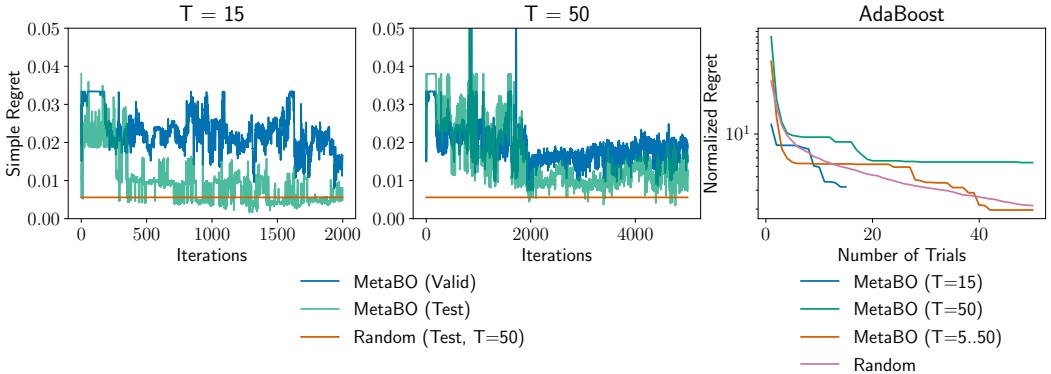

Figure 6: Left two plots: Training MetaBO for different number of trials $T$. Right plot: comparing the final policy against a random search. Results for $T = 15$ match the results reported by Volpp et al. (2020).

| Method | AdaBoost | | | GLMNet | | | SVM | | |
|---|---|---|---|---|---|---|---|---|---|
| | 15 | 33 | 50 | 33 | 67 | 100 | 33 | 67 | 100 |
| FSBO | **3.10** | **1.13** | **0.80** | *0.42* | 0.22 | *0.16* | **0.79** | **0.51** | **0.36** |
| FSBO (REPTILE) | *3.80* | 2.13 | *1.21* | **0.36** | **0.19** | **0.14** | 1.48 | 0.89 | 0.72 |

Table 2: Comparing different model-agnostic meta-learning approaches. Again, the best results are in **bold** and results that are not significantly worse than the best are in *italics*.

hyperparameters. The kernel (linear, polynomial or RBF) and the continuous trade-off parameter $C \in [2^{-10}, 2^{10}]$ must always be selected. The degree $d \in \{2, 3, 4, 5\}$ is only considered for the polynomial kernel and the bandwidth $\gamma \in [2^{-10}, 2^{10}]$ is only considered for the RBF kernel.

We further use the AdaBoost metadata which has been used to evaluate MetaBO (Volpp et al., 2020). According to Wistuba et al. (2015b), this metadata was created using AdaBoost (Kégl & Busa-Fekete, 2009) with decision products as weak learners. The authors designed a grid over the two continuous hyperparameters: the number of iterations and the number of product terms. The number of iterations was chosen from $\{2, 5, 10, 20, 50, 100, 200, 500, 10^3, 2 \cdot 10^3, 5 \cdot 10^3, 10^4\}$, the number of product term was chosen from $\{2, 3, 4, 5, 7, 10, 15, 20, 30\}$.

The metadata was created by running a grid search on 50 different classification data sets, using classification accuracy as an objective. Compared to the other two metadata sets, this one is significantly smaller. Its use is mainly motivated by comparing to MetaBO under the same circumstances as used to evaluate MetaBO. For that reason, we do not use the leave-one-data-set-out cross-validation protocol but instead use the same fixed split created by Volpp et al. (2020).

Statistics about these metadata sets are provided in Table 3.

## D    META-LEARNING WITH REPTILE

In principle our few-shot surrogate can be combined with any model-agnostic meta-learning approach. In this section we compare our proposed meta-learning approach against Reptile (Nichol et al., 2018), a first-order approximation of MAML. We use the same hyperparameters and data augmentation as described in Algorithm 1. Reptile introduces a new hyperparameter, an outer learning rate. We set it to 0.1 and decay it linearly to 0. As summarized in Table 2, FSBO with the meta-learning approach described in Algorithm 1 is either better or not significantly worse than its variation with Reptile.

| **AdaBoost** | | **GLMNet** | | **SVM** | |
|---|---|---|---|---|---|
| Data set | Runs | OpenML ID | Runs | OpenML ID | Runs |
| A9A | 108 | 3 | 12796 | 37 | 2429 |
| W8A | 108 | 31 | 60721 | 3485 | 1000 |
| ABALONE | 108 | 37 | 17950 | 3492 | 3217 |
| APPENDICITIS | 108 | 219 | 13963 | 3493 | 1828 |
| AUSTRALIAN | 108 | 3492 | 16968 | 3494 | 1854 |
| AUTOMOBILE | 108 | 3493 | 19931 | 3889 | 1951 |
| BANANA | 108 | 3889 | 13942 | 3891 | 3083 |
| BANDS | 108 | 3891 | 13917 | 3899 | 2487 |
| BREAST-CANCER | 108 | 3899 | 15954 | 3902 | 1105 |
| BUPA | 108 | 3903 | 39155 | 3903 | 1018 |
| CAR | 108 | 3913 | 22949 | 3913 | 1448 |
| CHESS | 108 | 3917 | 16960 | 3918 | 1459 |
| COD-RNA | 108 | 3918 | 20984 | 3950 | 1998 |
| COIL2000 | 108 | 9914 | 37635 | 6566 | 2702 |
| COLON-CANCER | 108 | 9946 | 25532 | 9889 | 2485 |
| CRX | 108 | 9952 | 25854 | 9914 | 1498 |
| DIABETES | 108 | 9967 | 21994 | 9946 | 1291 |
| ECOLI | 108 | 9978 | 19956 | 9952 | 2483 |
| GERMAN-NUMER | 108 | 9980 | 31961 | 9967 | 1492 |
| HABERMAN | 108 | 9983 | 37134 | 9971 | 1381 |
| HOUSEVOTES | 108 | 10101 | 66277 | 9976 | 1473 |
| IJCNN1 | 108 | 125923 | 35555 | 9978 | 1503 |
| KR-VS-K | 108 | 145847 | 29927 | 9980 | 2943 |
| LED7DIGIT | 108 | 145857 | 31180 | 9983 | 987 |
| LETTER | 108 | 145862 | 12954 | 10101 | 1871 |
| LYMPHOGRAPHY | 108 | 145872 | 17835 | 14951 | 1765 |
| MAGIC | 108 | 145953 | 47439 | 34536 | 1499 |
| MONK-2 | 108 | 145972 | 19972 | 34537 | 1489 |
| PENDIGITS | 108 | 145979 | 12979 | 145878 | 1244 |
| PHONEME | 108 | 146064 | 43785 | 146064 | 1435 |
| PIMA | 108 | | | | |
| RING | 108 | | | | |
| SAHEART | 108 | | | | |
| SEGMENT | 108 | | | | |
| SEISMIC | 108 | | | | |
| SHUTTLE | 108 | | | | |
| SONAR-SCALE | 108 | | | | |
| SPAMBASE | 108 | | | | |
| SPECTFHEART | 108 | | | | |
| SPLICE | 108 | | | | |
| TIC-TAC-TOE | 108 | | | | |
| TITANIC | 108 | | | | |
| TWONORM | 108 | | | | |
| USPS | 108 | | | | |
| VEHICLE | 108 | | | | |
| WDBC | 108 | | | | |
| WINE | 108 | | | | |
| WINEQUALITY-RED | 108 | | | | |
| WISCONSIN | 108 | | | | |
| YEAST | 108 | | | | |
| TOTAL | 5400 | TOTAL | 804159 | TOTAL | 54418 |

Table 3: Metadata set statistics. OpenML ID refers to a task on openml.org.

