# OpenReview forum: "Few-Shot Bayesian Optimization with Deep Kernel Surrogates"
_ICLR.cc/2021/Conference — ICLR 2021 Poster_

### Official Review · AnonReviewer3 · 2020-10-27
**Novelty is unclear and description is inconsistent**

**Rating:** 5
**Confidence:** 4

**Review:**

This paper aims to solve the expensive Bayesian optimization from the view of few-shot learning by resorting to the usage of deep kernel learning for hyperparameter optimization. The major concerns regarding this paper are listed as below.

The description of GP in sec. 2.2 is confusing. The paper considers the noise variance \sigma_n^2 for K_n while ignoring it for K_* in eq. 5, which is inconsistent. The authors are recommended to refer to the GP model described in the book Gaussian Processes for Machine Learning by Rasmussen.

The second para of sec. 3 defines T source tasks, leaving no symbols for indicating the interested target task. Similarly, the reviewer cannot find the definition of target task in the following equations.

The paper states that the parameters \theta and w of the deep kernel are task-independent, and the task-dependent parameters are kept separate. Where is the definition of the task-dependent parameters? The authors should make it clear.

The reviewer did not clearly found the novelty of this paper. How to transfer knowledge from source tasks (simply sharing the parameters of deep kernel?), and what is the difference to existing transfer learning works in the studied few-shot Bayesian optimization framework?

---

> ### Author Response · Authors · 2020-11-19
> **Clarification of Novelty and Updated Description**
>
> ### GP Notation
> We have completely revised the GP section and now use a notation that comes closer to that used by Rasmussen. We would like to point out that the noise variable is only added the the diagonal of the matrix K. Contrary to your comment, it does not appear in K_* (we refer you to [Rasmussen's book, page 16, Equation 2.21](http://www.gaussianprocess.org/gpml/chapters/RW.pdf))
>
> ### Symbol for Target Task
> We have updated the manuscript and introduced the symbol T + 1 as the symbol for the target task.
>
> ### Clarification on Parameter Sharing
> All task-dependent variables were marginalized when deriving the final model. The final model therefore no longer contains any task-dependent parameters and only depends on the task-independent parameters w and theta. In this work we do not go into details, but refer to [M. Patacchiola et al.: Bayesian Meta-Learning for the Few-Shot Setting via Deep Kernels](https://papers.nips.cc/paper/2020/file/b9cfe8b6042cf759dc4c0cccb27a6737-Paper.pdf).
>
> ### Key Contributions
> We have described our contributions in detail at the end of the introduction and revised them again in order to highlight them more clearly. We believe that of all the contributions in our work, the following are the most important:
> 1. This is the first paper that, in the context of hyperparameter optimization (HPO), trains the initialization of the parameters of a probabilistic surrogate model (i.e. a deep kernel GP) from a collection of meta-tasks by few-shot learning and then transfers it by fine-tuning the initialized kernel parameters to a target task.
> 2. We are the first to consider transfer learning in HPO as a few-shot learning task.
> 3. We are also the first to propose using Deep Kernel GP as a shared surrogate network.
> 4. We set the new state of the art in transfer learning for the HPO and provide ample evidence that we outperform strong baselines published at ICLR and NeurIPS with a statistically significant margin.
>
> ### Conclusion
> We believe we have addressed all of your points above and we would love to answer any additional questions you may have.

---

### Official Review · AnonReviewer2 · 2020-10-27
**Technical contribution is incremental**

**Rating:** 4
**Confidence:** 5

**Review:**

This paper proposes to simply learn/optimize the deep kernel parameters and hyperparameters of the GP using the data from all tasks (equation 9) and use such a deep GP kernel for BO. Instead of maximizing the log marginal likelihood of a single dataset (as is typically the case for a learning task), they propose to maximize the sum of log marginal likelihoods over the datasets of all tasks, which I view to be an incremental technical contribution. This paper is not about innovations in the acquisition function in BO. Their proposed approach outperforms the tested methods on 3 benchmark datasets.

A drawback of their proposed method (equation 9) is that in contrast to some existing meta-BO algorithms, it cannot exploit the GP posterior means and variances when such information is available from previous BO tasks.

Considering the diversity of the types of datasets in the AdaBoost experiment, can the authors give an interpretation of the max likelihood estimates of theta and w in equations 8 and 9 in the context of this experiment?

For the GLMNet and SVM experiments, would it be possible to instead combine all the datasets over tasks and construct a *joint* likelihood over them in equation 9 instead of a sum of likelihoods over tasks? How would the results differ in this case?

For the experiments conducted, large amounts of data are drawn from many available previous BO tasks, especially for GLMNet and SVM. This seems to be in disagreement with the setting of BO where the unknown objective functions are expected to be costly to evaluate (e.g., in hyperparameter optimization). In the context of BO, it would be meaningful to consider the practical setting where only small amounts of data are available from a few expensive BO tasks. In this case, how would the proposed approach perform compared to the tested methods?

An empirical comparison with the state-of-the-art meta-BO algorithm: weighted GP ensembles (Feurer et al. 2018) should be included.

From the results presented in Table 1 and Fig. 2, it does not seem like only a few shots/BO iterations are needed for the experiments performed in this paper. How do the results compare when only a few shots are used?

Equations 9 and 10: Shouldn't the log marginal likelihood be over y's instead of f's?

Equation 13: Exactly how is the loss function L(f^(t),X) defined? The author mentions that it is a normalized regret. The exact expression is needed here. In particular, I have noticed that f^(t) is not in bold: does this mean that only 1 "test" point is selected per task t?

Page 6: Can the authors provide the exact details on "the settings being sampled in proportion to their performance according to the predictions"?

The authors can consider doing experiments on hyperparameter optimization of larger-scale CNNs, which is commonly seen in BO works.


Minor issues

Page 3: adaption or adaptation

The following reference would be relevant to the context of meta/transfer BO:

Z. Dai, B. K. H. Low, and P. Jaillet (2020). Federated Bayesian optimization via Thompson sampling. In Proc. NeurIPS.

---

> ### Author Response · Authors · 2020-11-20
> **Scientific Contribution is not Incremental**
>
> ### Technical Contribution is Incremental
> From an implementation point of view, this method could be very similar to other methods proposed in this area. However, we are not suggesting an implementation hack to make Bayesian optimization more efficient. Instead, we propose to take recently proposed few-shot learning techniques and apply them to surrogate models used in Bayesian optimization.  This is a new idea and we are the first to provide empirical evidence that this is a very promising idea that is worth investigating further. The fact that this is relatively easy to implement is just another plus point.
>
> ### Alleged Drawback
> It is true that some methods follow a two-step approach that combines posterior means and covariances of different tasks. However, we would like to point out that these posteriors are derived from the data, so it is unclear why this approach would be superior to ours. In our approach we determine the posterior of all tasks directly by means of an end-to-end optimization without the intermediate step of RGPE, in which per-task GPs must first be trained and then aggregated. Please also note that our GP is based on a parametric kernel, which was initialized from the solution of the meta-learning optimization on the source tasks, but is still fine-tuned on the target task. Overall, we see end-to-end learning of the joint initialization of the shared kernel parameters as a different paradigm than RGPE. We added RGPE as a baseline and found that our approach not only scales better, but also outperforms it with statistical significance. Our implementation of RGPE is based on the code provided [here](https://botorch.org/tutorials/meta_learning_with_rgpe) since no official code is available.
>
> ### Interpretation of the Maximum Likelihood Estimates
> The maximum likelihood estimates provide a model with good generalization capabilities. The initial estimates allow a quick adaptation to new unseen tasks. Please take a look at our motivational example in section 5. In this case, each task is a completely different sine wave. The expected value for each argument x is 0. A trivial approach that would learn directly from all of these data points would not determine initial estimates that provide a model with good generalization capabilities.
>
> ### Joint Likelihood
> Our proposed generative model already uses a joint likelihood for different tasks. We assume that you propose to model the dependency between tasks differently. You are likely proposing to investigate what would happen if all the observations were assumed to be jointly Gaussian. Note that this approach is different from ours in that it ignores the fact that tasks are different, which is an important assumption.
>
> ### Disagreement with the Setting of BO
> We would like to point out that our protocol exactly follows the setup of the baselines, namely MetaBO and ABLR, which were published at ICLR 2020 and NeurIPS 2018. These state-of-the-art papers represent the gold standard in terms of empirical protocol and we followed this standard rigorously.
>
> ### Method Require More Than Few Shots
> Of course it is desirable to solve every problem in a few shots, but it is also unrealistic. In the case of Few-Shot Bayesian optimization, we are referring to Bayesian optimization, in which a few-shot surrogate model is used. However, this does not mean that we can guarantee that every problem can be solved within a few tries. We would like to point out that we still achieve significant improvements compared to the state of the art and consider the idea of few-shot learning in the context of hyperparameter optimization as a new and interesting direction. We refer to the appendix for results with fewer tries.
>
> ### Definition and Notation
> Thank you for pointing out these various minor issues. We have adapted our latest version accordingly.
>
> ### Conclusion
> We believe we have addressed all of your points above and we would love to answer any additional questions you may have.

---

### Official Review · AnonReviewer1 · 2020-10-29
**A nice application of deep kernel transfer in hyperparameter optimization.**

**Rating:** 6
**Confidence:** 3

**Review:**

Update: I appreciate the response to address my concerns carefully. My major concerns were the lack of novelty and some unclear descriptions on details in methods and experiments, but most of them have been well addressed. At first, I didn't see the technical challenges when applying DKL or any multi-task GPs into hyperparamter optimization. After reading the response and the manuscript again, I'm convinced the task augmentation plays a critical role in this setting. And, additional information from the revised manuscript helps to understand the details in method and experiments. So, I increased my score by two points.

**Strengths**
Solving few-shot regression tasks is interesting for warm-starting Bayesian optimization.

**Weaknesses**
1. Technical novelty seems to be weak. I have some concerns on the proposed approach.
2. Compared to baselines, the experiments are not strong.
3. The presentation needs to be improved.

**Major comments**

“A deep kernel ϕ is used to learn parameters across tasks such that all its parameters θ and w are task-independent. All task-dependent parameters are kept separate which allows to marginalize its corresponding variable out when solely optimizing for the task-independent parameters.” -> It didn’t clearly state which parameters are task-dependent and which are not. \theta and w are the only learnable parameters by maximizing a marginal log-likelihood. So, which parameters are task-dependent?

Compared to few-shot regression in a meta-learning framework, I didn’t find the benefits of the proposed approach. In addition, the proposed method doesn’t have a regularzing effects for few-shot tasks, since there is no meta-learning phase and no parameters shared across tasks.

“In hyperparameter optimization, we are only interested in the hyperparameter setting that works best according to a predefined metric of interest. Therefore, only the ranking of the hyperparameter settings is important, not the actual value of the metric.” -> If we discretize all continuous hyperparameters, this argument makes sense. But, I couldn’t find the hyperparameter setting for AdaBoos, GLMNet, and SVM in the paper. How many discrete/continuous hyperparameters? And, how to determine the range for each hyperparameter? Did you follow the same setting in this paper? Initializing Bayesian Hyperparameter Optimization via Meta-Learning, M. Feurer et al. AAAI’15.

---

> ### Author Response · Authors · 2020-11-19
> **Important Point Criticized are the Result of Misunderstandings**
>
> ### Compared to baselines, the experiments are not strong.
> We compare our methodology to the latest methods in the field (MetaBO (ICLR 2019) and ABLR (NeurIPS 2018)).
> We consider three different real-world optimization problems, which is more than is considered in other works (MetaBO and ABLR both only evaluate on two real-world problems).
> In addition, we would like to emphasize that these three different problems are aggregated results of 75 different optimization problems, for which we report statistically significant improvements compared to the state of the art (leave-one-dataset-out cross-validation for SVM and GLMNet leads to 30 + 30 = 60 optimization problems, the MetaBO split for AdaBoost leads to 15 further optimization problems).
> Would you mind explaining what you refer to when you say that our "experiments are not strong"? What is it that is missing and you would like to see? What would make the experiments stronger?
>
> ### Clarification on Parameter Sharing and Meta-Learning
> Our final model contains only the parameters w and theta, which are both task independent. Our model therefore clearly contains parameters which are shared across tasks and we also use a meta-learning phase. Referring to Algorithm 1, you can clearly see that all parameters (w and theta) are updated regardless of which task is selected. Apart from w and theta there are no further parameters.
>
> "Therefore, only the ranking of the hyperparameter settings is important, not the actual value of the metric."
> This sentence suggests that it is more important to correctly reflect the relative rankings of the response function values (e.g., validation errors) between pairs of hyperparameters than to actually correctly predict the response function value. In other words, we want to know if one hyperparameter setting is better than another, rather than estimate the exact validation error. We emphasize that the task of hyperparameter optimization (HPO) is not to estimate the validation error of a hyperparameter setting (i.e. not a regression task). That is, we are interested in the hyperparameter setting that will achieve the lowest validation error, but not the error itself. To conclude this point, we emphasize that this is a standard definition of the HPO problem.
>
> ### Metadata Details
> At your request, we have expanded Section C in the Appendix and provided further details on the metadata, including hyperparameter ranges and number of continuous and discrete hyperparameters.
>
> ### Conclusion
> We believe that the most important points criticized here are the result of misunderstandings.

---

### Official Review · AnonReviewer4 · 2020-10-29
**A new approach for Hyperparameter Optimization using Few-Shot learning - Good empirical results but novelty over prior work does not appear significant**

**Rating:** 6
**Confidence:** 4

**Review:**

**Quality and Clarity**

 The authors clearly describe the problem and the proposed solution. The explanation in some parts can be improved (see Queries and Suggestions below) but overall the paper is well written.

**Originality and Significance**

While the problem of hyperparameter optimization is extremely important and well studied, the main contribution of this work appear to be some modifications to the transfer learning based approaches that leverage the performance of hyperparameter settings on related tasks to tune the hyperparameters for a new task. The techniques like Warm Start, Data Augmentation, and sharing of NN and GP parameters across all tasks are modifications that can potentially improve the practical performance of many approaches in this space and thus positioning this work as a set of techniques that can be applied to multiple models in this space might make it more impactful than the current approach of positioning it as a single model that can outperform other models.

**Strengths**

1. The idea of using task independent parameters for both the neural network and the GP appears to be novel and enables the model to learn from the data across all the old tasks and the initial data from the new task.

2. The data augmentation and warm start approaches are intuitive and appear to give clear gains in practice.

**Weaknesses**

1. I do not think it is correct to apply mini-batch SGD to the task-wise losses in the manner of Algorithm 1, since the individual losses (GP log-likelihood) will not be additive over the samples of a given task and so the gradient estimates from the mini-batch would be biased. It might work in practice but I would appreciate some discussion/clarification on this in the paper.

2. While using mini-batches is based on the rationale that the number of samples for each task might be large, the data augmentation approach proposed seems to be intended for the setting when "only a few examples are available". Thus it is not clear if the targeted setting involves few or many examples per task.

3. The data augmentation and warm start approaches are largely heuristic and it is not clear whether they will always be beneficial.

**Queries/Suggestions**

1. Please explain the mutation and crossover operations in the Warm Start approach more clearly (preferably with an example). I am not entirely clear on what they are doing.

2. Please provide a mathematical formulation of the normalized regret loss used to evaluate the methods in the Experiments.

3. Can the Warm-Start and fine-tuning steps be performed with the ABLR baseline as they have been performed with the Multi-Head GP baseline? If so, please include those results as well.

4. Please include some discussion/clarification on the points 1 and 2 under Weaknesses above.

**Comments after Author Response**

I thank the authors for their response. The explanation of the main approach has certainly improved and the details of task augmentation and warm start are much clearer now. I also appreciate the added discussion on bias in Stochastic Gradients for GP training. The reference cited does seem to indicate that the training will converge despite bias in gradients. The warm start and task augmentation approaches still seem a bit heuristic to me. The task augmentation approach seems to assume an inherent linearity in errors/noise for the metric of interest which may not hold in practice. The choice of loss function for warm start also seems rather arbitrary. It is not entirely clear why choosing $\mathbf{X}^{\text{(init)}}$ that minimizes (12) is a good idea. However as my main concern about the validity of SGD in training the model has been addressed and both the task augmentation and warm start approaches seem somewhat intuitive, at least at the idea level, I am increasing my score by one point.

---

> ### Author Response · Authors · 2020-11-19
> **Addressing your concerns and clarifying some of your interpretations**
>
> ### Weakness 1: Correctness of Training Protocol
> In our work, we do not follow any unusual training protocol, but follow the procedure that is also used in many similar few-shot algorithms. At this point we would like to refer to a paper recently accepted on NeurIPS 2020 that uses the same training protocol: [M. Patacchiola et al.: Bayesian Meta-Learning for the Few-Shot Setting via Deep Kernels](https://papers.nips.cc/paper/2020/file/b9cfe8b6042cf759dc4c0cccb27a6737-Paper.pdf).
> We understand that SGD uses noisy approximations of the true gradient, but it turned out to be quite useful. Can you clarify what you think the gradients are biased for so we can address your concern?
>
> ### Weakness 2: Motivation for Task Augmentation
> The task augmentation strategy is not used to generate additional examples for cases with few examples. Instead, it is used to generate more tasks from the existing ones. These newly generated tasks differ from the original only in the label. The labels are re-scaled to a random range. It is important here that the values of the labels change, but their ranking remains unchanged. The motivation behind this approach is to learn a representation that is invariant to different scales of the labels.
> Since the trained model is thus invariant on the label scale, no label normalization is required for the target task.
> In summary, the task augmentation is only applied to the source tasks (for which we have relatively many examples) but not to the target task (for which we have few examples), since the purpose of the task augmentation is not to generate more training data for the few-shot task.
> We have made this clearer by adding two sentences to the manuscript.
>
> ### Weakness 3: How well do Task Augmentation and Warm Starting Heuristics Generalize?
> In our work we empirically demonstrate that both heuristics offer an advantage.
> Not only do we compare our methodology to the latest methods in the field (MetaBO (ICLR 2019) and ABLR (NeurIPS 2018)), we also thoroughly examine the effects of the heuristics in our ablation study to provide empirical evidence that they are beneficial.
> We consider three different real-world optimization problems, which is more than is considered in other works (MetaBO and ABLR both only evaluate on two real-world problems).
> In addition, we would like to emphasize that these three different problems are aggregated results of 75 different optimization problems, for which we report statistically significant improvements compared to the state of the art (leave-one-dataset-out cross-validation for SVM and GLMNet leads to 30 + 30 = 60 optimization problems, the MetaBO split for AdaBoost leads to 15 further optimization problems).
> In conclusion, we believe we are providing compelling empirical evidence showing that our heuristics generalize.
>
> ### Query 1: Mutation and Crossover Example
> We have added examples to the manuscript (Figure 2) to help understand this method.
>
> ### Query 2: Definition of Normalized Regret
> We added this definition (Equation 15).
>
> ### Query 3: Additional Baseline - ABLR + WS
> ABLR already uses fine-tuning. We added the results for ABLR + WS to Table 1.
>
> ### Query 4: Additional Clarifications
> As mentioned above, we have already addressed some of your points and updated the manuscript. We ask for your clarification on the others.
>
> ### Application to Other Approaches
> You suggest applying our idea to multiple approaches rather than introducing a new approach. That sounds like an interesting and promising idea and we are excited to try it. However, it is not clear what approaches you have in mind. Most of the methods are based on GPs that we have already covered in our work. There are some methods that are based on Bayesian neural networks. Since our method could also be interpreted as a Bayesian neural network, we have covered this as well. We look forward to your reply.
>
> Even though we do not yet understand how to implement your proposed idea, we believe that our propose method is novel. The idea to use few-shot learning techniques in the scope of Bayesian optimization is novel. Of course that involves parameter sharing which has been done before but as you pointed out, we are able to show that we outperform these methods.
>
> ### Conclusion
> We believe we have addressed all of your points above and we would love to answer any additional questions you may have.

---

> > ### Comment · AnonReviewer4 · 2020-11-23
> > **A Couple of Clarifications**
> >
> > Thank you for your responses. I will take them into consideration when deciding my final score. Let me try to clarify two points from my review which seemed to have caused some confusion:
> >
> > 1. The issue of biased gradients will arise because considering only a batch of data (even if the batches are i.i.d) for a given task will lead to a biased estimate of the GP negative log likelihood of that task. This is because the GP negative log-likelihood for a task is not additive over the data samples  for that task (unlike loss functions like least squares or cross entropy). See for eg. the discussion here https://stats.stackexchange.com/questions/364293/is-stochastic-gradient-descent-biased . I believe biased gradients often work well enough in practice so it is okay to use them but, if the gradients are indeed biased as I suspect, it would be a good idea to mention it since works like [1] have seen improvement on reducing the bias. The specific approach for reducing the bias would depend on the loss function but mentioning it here can spur future work in that direction for this loss functions.
> >
> > 2. Regarding my comment about applying the ideas to other approaches, I believe the Warm Start idea has already been applied to other approaches (GP, ABLR and Multi-head GPs). In the same vein I was wondering if the Task Augmentation idea could also be used across the board for different approaches wherever hyper parameter optimization is viewed in a transfer learning framework. I do not expect you to implement it, but it might be worth mentioning in the paper if it is something that could be studied in future work or implemented by practitioners.
> >
> > [1] Belghazi, Mohamed Ishmael, et al. "Mine: mutual information neural estimation." arXiv preprint arXiv:1801.04062 (2018).

---

> > > ### Author Response · Authors · 2020-11-24
> > > **Answer After Clarifications**
> > >
> > > Thanks for the clarifications. We would like to answer your comments.
> > >
> > > 1. We now understand what you are referring to. We have updated the manuscript and dedicated a paragraph to this aspect. We mainly refer to a [NeurIPS 2020 paper](https://papers.nips.cc/paper/2020/hash/1cb524b5a3f3f82be4a7d954063c07e2-Abstract.html) [1] that examines this issue. This paper shows that an optimization with a stochastic gradient descent for Gaussian processes not only works in practice, but the authors prove that one can guarantee that this method also converges and that the model parameters are correctly determined.
> > > 2. We already discuss this in our work (below equation 11). The reason it doesn't work for the existing methods depends on the existing method itself. Some of them learn a standard GP for the entire data of all tasks and differentiate tasks based on metafeatures. These approaches cannot be scaled well even for small amounts of data, so that the generation of further data is hopeless. In addition, it is unclear how metafeatures should be generated. Other methods like ABLR include task-dependent parts in their modeling. This means not only that these models grow with each new task generated, but also that each of these task-dependent parameters is only trained for a single batch, which is insufficient. To sum up, there is no easy solution to add this idea to existing solutions. With our work we propose a new method that can use task augmentation.
> > >
> > > [1] Hao Chen, Lili Zheng, Raed AL Kontar, Garvesh Raskutti: Stochastic Gradient Descent in Correlated Settings: A Study on Gaussian Processes, NeurIPS 2020.

---

> > > > ### Comment · AnonReviewer4 · 2020-11-24
> > > > **Thank you**
> > > >
> > > > Thank you for updating the paper and sharing the NeurIPS 2020 paper on this issue. I was not familiar with that work. I will take these updates into consideration when determining my final score.

---

### Decision · Program_Chairs · 2021-01-07
**Final Decision**

**Decision:**

Accept (Poster)

**Comment:**

This paper uses deep kernel learning to develop a compelling framework for hyperparameter optimization in a few-shot setting, with empirically strong results. Please carefully account for all reviewer comments in the final version.